# Characterization of Al-Containing Industrial Residues in the ESEE Region Supporting Circular Economy and the EU Green Deal

**DOI:** 10.3390/ma17246245

**Published:** 2024-12-20

**Authors:** Emilija Fidanchevski, Katarina Šter, Maruša Mrak, Milica Rajacic, Bence David Koszo, Andrej Ipavec, Klemen Teran, Gorazd Žibret, Vojo Jovanov, Nikolina Stamatovska Aluloska, Mojca Loncnar, Lea Žibret, Sabina Dolenec

**Affiliations:** 1Faculty of Technology and Metallurgy, Ss. Cyril and Methodius University in Skopje, 1000 Skopje, North Macedonia; emilijaf@tmf.ukim.edu.mk (E.F.); vojo@tmf.ukim.edu.mk (V.J.); 2Slovenian National Building and Civil Engineering Institute, 1000 Ljubljana, Slovenia; katarina.ster@zag.si (K.Š.); marusa.mrak@zag.si (M.M.); lea.zibret@zag.si (L.Ž.); sabina.dolenec@zag.si (S.D.); 3Radiation and Environment Protection Department, Vinca Institute for Nuclear Science, National Institute of the Republic of Serbia, University of Belgrade, 11000 Belgrade, Serbia; milica100@vin.bg.ac.rs; 4Bay Zoltán Nonproft Ltd. for Applied Research, H-6726 Szeged, Hungary; bence.koszo@bayzoltan.hu; 5Alpacem Cement, d.d., Anhovo 1, 5210 Deskle, Slovenia; andrej.ipavec@alpacem.si; 6Geological Survey of Slovenia, 1000 Ljubljana, Slovenia; klemen.teran@geo-zs.si; 7Cementarnica “USJE”AD Titan Group, 1000 Skopje, North Macedonia; nikolinas@usje.mk; 8SIJ Acroni d.o.o., Cesta Borisa Kidriča 44, 4270 Jesenice, Slovenia; mojca.loncnar@acroni.si; 9University of Ljubljana, Faculty of Natural Sciences and Engineering, Department of Geology, Aškerčeva ulica 12, 1000 Ljubljana, Slovenia

**Keywords:** Green Deal, Al-containing industrial residues, circular economy, fly ash, slag, red mud

## Abstract

The increase in industrial waste generation presents a global problem that is a consequence of the needs of modern society. To achieve the goals of the EU Green Deal and to promote the concept of circular economy (CE), the valorization of industrial residues as secondary raw materials offers a pathway to economic, environmental, energetic, and social sustainability. In this respect, Al-containing industrial residues from alumina processing (red mud), thermal power plants (fly ash and bottom ash), and metallurgy (slag), as well as other industries, present a valuable mineral resource which can be considered as secondary raw materials (SRMs) with the potential to be used in construction, supporting the concept of circular economy. This paper focuses on the characterization of 19 secondary raw materials from the East South-East Europe (ESEE) region regarding their physical, chemical, mineralogical, and radiological characteristics. The goal is to provide a foundation for future innovations based on secondary raw materials, in alignment with the EU Green Deal and the principles of circular economy. The results showed that fly ash has the potential to be the best material among those analyzed to be used in the cement industry, mainly due to its favorable radiological and mineralogical properties. However, it is important to control the amount of free lime in the mixture, ensuring it remains below 10%. After evaluating secondary mineral raw materials for metal recovery, the results indicate that these materials are not viable sources for base metals or other technology-critical metals, such as REEs.

## 1. Introduction

The EU Green Deal (EU GD) is a new growth strategy aimed at transforming the EU into a fair and prospective society. It focuses on creating a modern and competitive economy with the goal of completely reducing greenhouse gas emissions by 2050 in the whole European Union [1]. The mobilization of industries for a clean and circular economy is essential for achieving the new EU strategy. The circular economy action plan (ECAP) supports the transition from a linear model to a circular model, where waste becomes a valuable resource [2]. In addition, it recommends making use of mineral resources in more efficient way, i.e., to recycle and recover raw materials from any waste streams [3]. The concept of circular economy supports the utilization of secondary raw materials (SRMs), as these materials provide economic, energetic, and environmental benefits. Nonetheless, SRM still contains many valuable components and possesses characteristic physico-chemical properties for producing value-added products that, in recent years, have gained significant interest. In this regard, slag from the steel industry, red mud from alumina production, and fly ash and bottom ash from thermal power plants and the paper industry, among other wastes/by-products, present potential for valorizing as SRMs in cement plants [4] and in the production of bricks [5], glass–ceramics [6], adsorbents [7], zeolites [8], etc.

In 2021, the world iron slag production was estimated to be between 340 million and 410 million tons, and the steel slag production was estimated to be between 190 million and 280 million tons [9]. The most recent European statistics for 2020 indicate that 20.8 Mt of blast furnace slag (BFS) and 12.4 Mt of slag from steel production were produced in Europe [10]. The recycling of these slags provides a number of environmental benefits, including the preservation of natural resources, the recovery of valuable metals, and a reduction in the volume of solid waste. BFS is mainly used as a cement and concrete additive (approx. 85% of total BFS utilization); meanwhile, the utilization of steel slag is more diverse. It is estimated that 57% of the total steel slag is used in road construction, 25% for metallurgical use and internal storage, 5% as fertilizer, 4% for cement and concrete additives, and 2% for hydraulic engineering, while the rest is used for other purposes.

Red mud as a residue from the Bayer process (alumina production from bauxite ore) is estimated at 120–150 Mt of global annual production, and its accumulation is estimated at around 4 billion tones [11]. Red mud has been used for metal recovery, adsorption, soil amendment, in catalysis, and in oxidation reactions, as well as in the production of construction and building materials, but only in small portions [12].

According to the ECOBA (European Coal Combustion Product Association 2016), 15% and 9% of the total produced (40 million tons) coal combustion products (CCP) belong to fly ash and bottom ash, respectively. Fly ash has been widespread, used in cement and concrete production [13], bricks and blocks [14], glass–ceramics [15], etc. Bottom ash could be used as a supplementary material in cement or as aggregate in concrete [16]. Both fly ash and bottom ash can be used for innovative products such as aerogels, geopolymers, rare earth elements, zeolite, etc., supporting the green transition in construction [17].

Cement plants, as an energy-intensive industry, have invested serious efforts to achieve the goals defined in the EU GD and ECAP. Among many approaches towards decarbonization, utilizing secondary raw materials (SRMs) presents one of the ways to decrease the CO_2_ footprint [18]. SRM can be incorporated in cement production as a raw meal of clinker (in the first stage of cement production) or as supplementary cementitious material (at the later stage of production, acting as a hydraulic or mineral additive) [19]. The incorporation of secondary raw materials in belite–sulfoaluminate cement (BCSA) was recently investigated. Namely, the recent investigations showed that bottom ash [20] and fly ash from thermal power plants [21], as well as steel slag [4], could be successfully embedded in BCSA. There is still potential to explore the use of characterized secondary raw materials for BCSA production and other applications. For example, studies have shown that alumina-rich wastes can be effectively utilized in geopolymers, with alkali-activated slag and fly ash geopolymers emerging as innovative, environmentally friendly alternatives to conventional OPC, especially as fire-resistant alkali-activated cementitious materials [22,23].

The potential utilization of SRM relies on its chemical and mineralogical composition, particularly the amount of amorphous phase contained within, its granulometry, and the content of natural radionuclides which defines these SRMs as NORMs (Natural Occurring Radioactive Materials). The Council Directive 2013/59/EUROATOM, 2013 [24], Article 75, defines the norms for SRM to be assessed from a radiological point of view as they will be used in standard building practice.

Most of the secondary raw materials presented in this study were collected and analyzed within the RIS-ALiCE project. Many of them are also available in the RIS-ALiCE registry [25]. This project aimed to collect data on Al-containing residues from the East South-East Europe (ESEE) region and to evaluate their potential use in the production of low-CO_2_ mineral binders based on BCSA. This paper presents the results of the secondary raw materials characterized by their physical, chemical, mineralogical, and radiological aspects. The results present the basis for innovative solutions for utilizing secondary raw materials from the ESEE region, in line with the EU Green Deal and the circular economy model.

## 2. Materials and Methods

### 2.1. Materials

Approximately 50 kg of bulk composite sample was collected at each sampling site in the period between October 2019 and October 2020. Bulk samples were composed of a minimum of 10 subsamples of equal size sampled from randomly selected points of stockpiles or sedimentation ponds to achieve representatives of each sample. Distribution of the subsamples considered the shape and type of the residue deposit, possible gravitational segregation of the material, and size of the particles deposited.

Prior to further analyses, all collected samples (Figure 1) were air-dried at 20 °C to a constant mass, with humidity measurements taken both before and after drying. The only exception was the slag mineral residue (SL2) obtained from the processing of the mixture of EAF S slag and ladle slag. Due to its nature (wet sample), it was immediately dried after sampling at 105 °C to prevent phase transformation. For the reduction of the bulk sample (air dried), a coning and quartering protocol was used. The coning and quartering method was chosen to systematically reduce samples while preserving their representativeness. The process of mixing and quartering was repeated until the required size of the laboratory sample was obtained. Each homogenized sample was of such size that all the individual analyses could be carried out twice.

Table 1 contains basic data on 19 samples, which represent red mud from alumina production (1 sample), different slags from the steel industry (5 samples), fly and bottom ashes from thermal power plants (8 samples), fly and bottom ash from paper mills (2 samples), and other industrial residues (3 samples). Additionally, in the frame of the project, two fly ashes and bottom ash from the thermal power plant REK Bitola, Republic of North Macedonia [26], five fly ash samples from the thermal power plant Nikola Tesla, Serbia [27], and red mud from Alumina Zvornik, Bosnia and Hercegovina [28] were characterized, as well as query and mine waste [29]. The collected samples are shown in Figure 1.

### 2.2. Methods

The characterization of collected samples includes determination of the physical characteristics (moisture content, granulometry, BET specific surface area, particle density, and bulk density), chemical composition (main oxides and LOI at 950 °C, trace/rare earth elements (REEs)/heavy elements), total organic content (TOC), mineralogical composition (including amorphous and crystalline non-quantifiable phases (ACns)), and radiological characteristics.

### 2.3. Physical Characterization

Moisture content was determined on the as-received samples. Samples (~50–100 g) were dried at 105 °C to a constant mass. The moisture content (mc) was determined according to the difference between the wet mass and mass after drying by Equation (1):% mcwb = [(mw − md)/md] × 100(1)
where

mcwb is the moisture content of the sample;mw is the mass of the wet sample;md is the mass of the dry sample.

The particle density of samples (~20 g) was determined with the pycnometer method (helium pycnometer Quantachrome Ultrapyc 1200e, Anton Paar GmbH, Graz, AT) in accordance with the standard EN 1097-7, 2008 [30]. The bulk density of samples was measured according to standard JUSB.C8.023, 1982 [31]. The samples QS1 and CW1 were not analyzed due to the nature of the material (the samples were previously ground and therefore unsuitable for analysis). The specific surface area (SSA) was determined by the BET method according to the standard ISO 9277, 2010 [32] with Micromeritics ASAP-2020 (Norcross, GA, USA) using nitrogen adsorption at 77 K. The SSA was not determined for QS1 and CW1 due to the nature of the material (samples were previously ground and therefore unsuitable for analysis).

For all relevant samples, sieve analysis compliant with ISO 3310-2, 2013 [33] was performed using a HAVER EML digital plus device (test sieve shaker) and laboratory test sieves from ELE international (16 mm, 8 mm, 4 mm, 2 mm, 1 mm, 0.25 mm, 0.125 mm, 0.063 mm, and <0.063 mm). For the sieve analysis, we used 2.5 kg of individual representative samples. Some samples (FA1, FA2, FA3, QS1, and CW1) were not suitable for sieve analysis due to their nature (the samples in larger pieces were pre-ground, or all particles were smaller than 0.063 mm).

Particle size distribution (PSD), by laser granulometry, was determined for all samples. A Helos BR laser, by the manufacturer Sympatec (Clausthal-Zellerfeld, DE), was used for all samples except for SL3, SL4, PFA, PBA, QS1, WJ1, and CW1, for which a Microtrac SYNC Model 5001, Microtrac Retsch GmbH, Haan, DE was used. In total, ~100 g of dried samples was sieved through a 0.25 mm sieve prior to analysis. For the samples SL3, SL4, PFA, PBA, QS1, WJ1, and CW1, a 1 mm sieve was used. For PSD analysis, ~1 g of sample was loaded into the test cell and analyzed using a wet configuration in isopropanol. The sample QS1 had very fine particles, and therefore its PSD was determined using a wet configuration in demineralized water. SL5 was not suitable for laser granulometry due to its nature (the sample was in larger pieces that we had to pre-grind). Measurements were performed in parallel.

### 2.4. Chemical and Mineralogical Characterization

The main chemical oxides (SiO_2_, Al_2_O_3_, Fe_2_O_3_, CaO, SO_3_, MgO, Na_2_O, and K_2_O), Cl^−^, and loss of ignition at 950 °C (LOI) of samples were determined by wet chemistry according to EN 196-2 [34], while P_2_O_5_ and TiO_2_ were determined by X-ray fluorescence spectroscopy (XRF). For the XRF of most samples, WDXRF, Thermo Scientific ARL PERFORM’X, Thermo Fisher Scientific, Waltham, MA, USA; fused beads; and the UniQuant program were used, while for the samples SL3, SL4, PFA, PBA, QS1, WJ1, and CW1 S8, Tiger by BRUKER (Billerica, MA, USA), fused beads, and the “clinker” program were used. To prepare the fused beads, we used ~1 g of the previously ignited (950 °C) test sample. The samples were sieved and ground through a 90 µm sieve. The obtained sample was mixed with lithium tetraborate, which served as the flux, in a 1:10 ratio and afterwards, fused at 1100 °C to create beads.

Trace elements, rare earth elements (REEs), and heavy metals were determined by the ICP-ES/MS method (multi acid digestion, where the residue has been dissolved in HCl). Based on the Hungarian standard MSZ 525-17, 2013 [35], 0.5 g sample, 1.16 g lithium metaborate, and 0.05 g ammonium nitrate were fused in a platinum crucible at 1000 °C for 60 min. A total of 44 ml of 5% hydrochloric acid and 30 ml of distilled water were used to dissolve the glass bed, and the solution was filled up to 100.0 mL with distilled water and analyzed with ICP OES (Perkin Elmer Avio200, Waltham, MA, USA). The plasma conditions were as follows: 1500 W; 12 L/min plasma gas; 0.2 L/min aux gas; and 0.65 L/min nebulizer gas. The sample flow was 1 ml/min. A MiraMist nebulizer and baffled cyclonic spray chamber were used. Lutetium was used as an internal standard at a 1 mg/L level. For samples where precipitation occurred at the dissolution step, more hydrochloric acid was added to the solution. In some cases, the fusion was repeated with a 0.3 g sample to avoid the formation of the precipitate.

For determination of the total mercury content (in solids and liquids), the solid sample of 100 mg without pre-treatment or pre-concentration was used. The system used was a LECO 254 Advanced Mercury Analyser, AMA—Atomic Absorption Spectrometer, available from Leco, UK.

The presence of organic matter, expressed by the total organic content (TOC), was analyzed in all samples (~200 mg) by an Analizator CW-800M “Multiphase” (Lahr, DE), ELTRA using the dry incineration method, and the detection of products with an IR detector.

The mineralogical composition of the samples was determined using an X-ray diffractometer (PANalytical Empyrean, Malvern Panalytical, Malvern, UK) equipped with CuKα radiation and a PIXcel 1D detector (Malvern panalytical, Malvern, UK,). Samples (~10 g) were ground in an agate mortar to a particle size below 0.063 mm. The ground powders were manually back-loaded into circular sample holders (27 mm in diameter), and data for each sample were collected from 4° to 70° (2θ) using a step size of 0.026° (2θ) and a scan time of 197s. Samples were measured at 45 kV and 40 mA and rotated during data collection with a revolution time of 2 s. The amount of crystalline phase and ACn were estimated by Rietveld refinement using the external standard method (alumina powder, Al_2_O_3_; NIST SRM 676a) and the PANalyticalX’Pert High Score Plus diffraction software, version 4.9 (Malvern Panalytical, Malvern, UK), using the structures for the phases from the ICDD PDF 4+2016 RDB powder diffraction files.

### 2.5. Radiological Characterization

The activity of radionuclides in the samples was determined by the gamma spectrometry method. Measurements were performed on HPGe detectors (Canberra, Sturbridge, MA, USA), with relative efficiencies of 20%, 18%, and 50% according to the IAEA TRS 295 method [36]. Efficiency calibration was performed using a certified radioactive standard (1035-SE-40845-17 [37]) and secondary reference materials produced from a radioactive solution (1035-SE-40844-17 [38]) from the Czech Metrology Institute, which contained ^210^Pb, ^241^Am, ^57^Co, ^60^Co, ^137^Cs, ^139^Ce, ^85^Sr, ^109^Cd, ^88^I, and ^51^Cr, and can be traced back to the International Bureau of Weights and Measures—BIPM (Bureau International des Poids et Mesures). The radioactive materials used were of a similar density and packing geometry as the samples.

After preparation, the samples (125 mL and 250 mL) were sealed with beeswax in the measurement geometry and left for at least one month before measurement, in order to establish a radioactive equilibrium between radon and its progeny. The measurements lasted 60,000 s, and the spectra were analyzed with the GENIE 2000 software package.

It is recommended (EC, Radiation protection 112) that the radiological hazard controls associated with exposure to ^226^Ra, ^232^Th, and ^40^K be based on an effective dose of 1 mSv/year.

The annual effective dose rate (*E*) was calculated by Equation (2) [39], using a conversion coefficient of 0.7 Sv/Gy to convert the absorbed dose in the air into the effective dose in the human body. *D* is the absorbed dose in the air and (*p∙t*) is the annual exposure time, where *p* is the percentage of years during which humans are exposed to radiation (occupancy factor) and *t* is 8.760 h (number of hours in the year).
(2)$$E\;\left(mSv/y\right)=D\left(\frac{nGy}{h}\right)\cdot p\cdot t(h/y)\cdot 0.7\left(\frac{Sv}{Gy}\right)\cdot 10^{-6}$$

For estimating the outdoor effective dose rate (*E_out_*, *E_20%_),* the outdoor occupancy factor pout is 0.2, while for estimating the indoor effective dose rate (*E_ind_*, *E_80%_*), the calculation takes into account that people spend about 80% of time indoors (indoor occupancy factor *p_ind_* is 0.8) [39].

The absorbed dose in the air, *D* (nGy/h), is estimated based on Equation (3) [39] where *q_i_* is the specific dose rate for isotope “*i*” in (nGy/h)/(Bq/kg), and *A_i_* is the activity concentration of isotope “*i*” in Bq/kg:(3)$$D=q_{226_{Ra}}\cdot A_{226_{Ra}}+q_{232_{Th}}\cdot A_{232_{Th}}+q_{40_{K}}\cdot A_{40_{K}}$$

The value of qi is usually calculated by simulating different cases. For the external terrestrial gamma radiation absorbed dose rate (*D_terr_*) in the air at a height of 1 m above ground level, the *q_i_* for ^226^Ra, ^232^Th, and ^40^K are 0.462, 0.604, and 0.0417, respectively [39].

An Activity Concentration Index (*ACI*) is the most common screening method for assessing the dose caused by building materials. It is associated with gamma radiation exposure inside buildings that exceeds typical outdoor exposure, and is calculated using Equation (4) [40]. For superficial and other materials with restricted use, the dose criterion of 1 mSv is already satisfied at *ACI* < 6, but for materials used in bulk amounts, the value of *ACI* should be less than 1 [40].
(4)$$ACI=\frac{A_{226_{Ra}}}{300}+\frac{A_{232_{Th}}}{200}+\frac{q_{40_{K}}}{3000}$$

The hazard indices are a screening method for the dose caused by the use of certain materials. The external hazard index (*H_ext_*) reflects the external radiation hazard due to the emitted gamma radiation and it is calculated according to Equation (5) [41].
(5)$$H_{ext}=\frac{A_{226_{Ra}}}{370}+\frac{A_{232_{Th}}}{259}+\frac{q_{40_{K}}}{4180}$$

The value of this index should be less than 1 in order to keep the radiation hazard insignificant [41], and the value of *H_ext_* equal to 1 ensures *E_terr,80%_* is less than 1 mSv.

In addition to external radiation, radon and its short-lived products are also hazardous to respiratory organs, and that is quantified by the internal hazard index, *H_int_*, as in Equation (6) [41]. The value of *H_int_* should be less than 1 for material used indoors. Due to *H_int_* being stricter than *H_ext_*, an *H_int_* equal to 1 ensures an *E_terr,80%_* less than 1 mSv.
(6)$$H_{int}=\frac{A_{226_{Ra}}}{185}+\frac{A_{232_{Th}}}{259}+\frac{q_{40_{K}}}{4180}$$

The equivalent activity of radium (*Ra_eq_*) [41] is equal to 370∙Hext, so the criterion *Ra_eq_* < 370 Bq/kg is equivalent to the criterion *H_ext_* < 1. Also, the alpha index (*I_α_*) < 1 requires that *A*(^226^Ra) < 200 Bq/kg (*I_α_* = *A*(^226^Ra)/200) (Nordic 2000), which is a weaker criterion than *A*(^226^Ra) < 185 Bq/kg, which requires *H_int_* < 1. Due to the above, these parameters were not considered separately.

Although all the abovementioned parameters ensure the condition *E_terr,80%_*, for some cases they are too strictly defined (for example, if no one spends 80% of the hours in a year with the observed material). Therefore, these are only screening parameters, and for cases where their values are >1, it is necessary to estimate the dose for the situation in which the observed material is used.

To calculate the qi for estimating the absorbed dose of a building material, a standard model room [42] (a room of 20 m^2^ and 3 m in height, with concrete of 20 cm thickness as the construction material) is the most frequently used, in a case where all structures including the floor, walls, and ceiling (*D_all_*); the floor and walls (*D_fw_*); and only the floor (*D_f_*) were built with it, as well as a case where the material is used as the superficial material for all walls (*D_sup_*) [42]. To estimate the annual effective dose rate for a standard room, *p* = 0.8. The value of *q_i_* for approximated cases is used from Markkanen [42].

It should be noted that the dose reference value of 1 mSv/y refers to the excess gamma dose received outdoors, but the estimated absorbed dose in air based on Equation (3) (absorbed dose in air) is not an excess exposure to building materials, because concrete structures protect against gamma radiation from the undisturbed Earth’s crust. Using the average value of the absorbed dose in the air for the background, the excess dose rate in the room is therefore (*D*
*D_back_*) nGy/h. Therefore, the annual excess effective dose to the occupant is as follows: *E_exc_* = (*D*
*D_back_*) nGy/h 7000 h 0.7 Sv/Gy. An average outdoor background in Europe is 50 nGy/h [40]. To assess the potential health impact on the public due to exposure to the tested samples, the annual effective dose (*E*) of the total external absorbed gamma dose in air at a height of 1 m above ground level for outdoor and indoor cases and for four different standard rooms was calculated. Based on the results obtained for the observed materials, the activity concentration index (*ACI*), the external hazard index (*H_ext_*), and the internal hazard index (*H_int_*) were calculated.

## 3. Results and Discussion

### 3.1. The Results of Physical Characterization

The physical characteristics of the secondary raw materials are crucial parameters that determine their preparation processes before industrial use, including the need for drying, grinding, etc., which are heavily connected to energy consumption. The results of the measurements of the physical characteristics (moisture content, particle density, bulk density, and specific surface area) of the collected secondary raw materials are presented in Table 2.

The moisture content varies between 0 and 20.8 wt.%, and it is heavily influenced by the type of material. Generally, the red mud, slags, and fly ashes and bottom ashes from paper industry and thermal power plants were found to contain relatively small amounts of moisture (between 0 and 2.83 wt.%, except SL2). The samples WJ1 and CW1 have slightly higher moisture content, up to 8 wt.%. In the case of sample SL2, the increased moisture content was mainly because a wet process was used to extract metal prior to disposal. The by-product of quartz sand production (sand washing residues, QS1), contained the highest measured moisture (20.8 wt.%) in this study, which is probably heavily influenced by the treatment and storage processes at the producer of this secondary raw material.

The particle density (solid phase only, pores are excluded) of the sampled materials varies between 1.91 and 4.14 g/cm^3^, while the bulk density (including pores) varies between 478 and 2206 kg/m^3^. The highest particle density was determined for waste water jet sand (WJ1), while the lowest was for fly ash from the thermal power plant (FA3). The density of the supplementary cementitious materials (SCMs) is important information as it allows a more accurate calculation of the proportions of concrete mixtures by volume instead of by mass [43]. Comparing the average particle density of the collected secondary raw materials with those of soils, which is 2.65 g/cm^3^ [44], it can be concluded that the density of the materials from this study is in a similar size range to that of soils. Particle density comparison between the sampled materials and rocks revealed that the particle densities of slags and fly and bottom ashes from the paper industry are comparable to those of igneous rocks, the particle density of bottom ash from the thermal power plant is comparable to that of limestone, and the particle densities of red mud and fly ash from the thermal power plant are comparable to those of shales or sandstones [45]. Particle density plays a crucial role in optimizing concrete mixtures, particularly in terms of strength development and hydration, as it influences the packing density and performance of supplementary cementitious materials (SCMs) like fly ash, slag, and silica fume. It helps determine the volume and distribution of particles within the mix, enabling the fine-tuning of the particle size distribution (PSD) to improve its workability and hydration efficiency. High-density particles can effectively fill voids, enhancing concrete strength by minimizing excess pore spaces, while denser materials like slag and certain fly ashes reduce water demand, contributing to long-term durability [46,47,48].

The bulk density of soils varies between 1500 and 1700 kg/m^3^ [44], so it can be concluded that sampled materials (except WJI) are less dense than average soils for most of the materials. Only some slags and water jet sand (WJ1) have a higher bulk density than soils. The higher density of water jet sand (WJ1) in comparison to soil is due to the presence of garnets. The EAF slag, formed largely by a content of oxides of iron, has high density and low porosity [49]. The bulk density of supplementary cementitious materials (SCMs) is critical for optimizing concrete mix design as it influences packing density, reduces voids, and improves both the strength and durability of the material. High-density SCMs enhance workability and lower water demand, creating a cohesive mix while also supporting sustainability by enabling greater clinker replacement and reducing CO_2_ emissions during production. Proper understanding and control of SCM bulk density ensure precise mix proportioning, minimizing unreacted materials and maximizing performance [50,51,52].

The specific surface areas (SSAs) of the collected samples vary much more than the other measured physical properties. The largest SSA was measured for fly ash from the paper industry (PFA) and exceeds the average SSA values of red mud 4-fold, the average SSA of fly ash from the thermal power plant 7-fold, the average SSA of bottom ash 10-fold, the average SSA of slag 28-fold, and the average waste water jet sand (WJ1) SSA 68-fold. Significant variations of SSA were also observed within individual types of secondary raw material samples. It looks like specific conditions during the process of generation of secondary raw materials play a crucial role in the value of the SSA parameter, which cannot be generalized but shall be measured individually. The specific surface area (SSA) of cement plays a critical role in determining the water demand during cement hydration. As the SSA increases (such as with finer cement particles), the surface area available for reaction with water also increases, which, in turn, raises the water demand for a given mass of cement. This is because more water is required to wet the finer particles and facilitate the hydration process. Studies show that the water demand for cement rises significantly with finer grinding, which increases SSA. This effect is closely linked to the particle size distribution (PSD), as finer particles tend to compact more densely, enhancing hydration but requiring more water [53]. Additionally, the inclusion of supplementary cementitious materials (SCMs), such as fly ash or slag, often changes the particle size and SSA of the binder system, affecting both the packing density and water demand. For example, increased use of SCMs typically leads to a higher SSA, which requires more water for the mixture to achieve the desired fluidity [54]. In practical terms, controlling the SSA through particle size optimization can help manage water demand while balancing the fluidity and strength of cement-based mixtures [53]. Finer particles, which have a larger surface area, generally exhibit higher reactivity. This is due to the increased surface contact between the cement particles and water, enhancing the hydration process. Studies on various types of cement have shown that the SSA correlates with compressive strength, with finer cement (higher SSA) often leading to stronger concrete due to better hydration and chemical activity [55].

The granulometric composition of samples determined by sieve analysis is presented in Figure 2.

RM contained particles in size between 1 and <0.063 mm, while most of them were less than 0.25 and 0.125 mm. In the slag, the smallest particles (<0.125 mm) were in SL2 (mineral product) and SL4 (ladle slag), where 70.2 wt.% were less than 0.063 mm. The largest particles were in SL3 (EAF C slag), where most of them passed the 8, 4, 2, and 1 mm sieves. SL1 (EAF C slag) and SL5 (blast furnace slag) have the most particles between 1 and 0.5 mm. In PFA (fly ash from the paper industry), 79.0 wt.% of the particles were less than 0.063 mm. In FA6 and FA5 (fly ashes from thermal power plant), the particles were a bit larger—most of them were less than 0.25 and 0.063 mm. In PBA (bottom ash from the paper industry), the most particles passed sieves between 1 and 0.063 mm. In BA1 (bottom ash from the thermal power plant), the most particles passed sieves between 2 and 0.25 mm, while in BA2, 68.1 wt.% were less than 0.5 mm. In sample WJ1 (waste water jet sand), most of the particles passed sieves of 0.125 and 0.063 mm.

The results of the PSD (Figure 3) showed that most of the samples of secondary raw materials have a unimodal distribution, while the samples of RM, FA1, FA2, and FA6 have a bimodal distribution. According to the characteristic values D_10_, D_50_, and D_90_ (Figure 4), the largest mean diameter, D_50_, occurs in the sample of WJ1 (138.3 µm), while the smallest is for the red mud. In general, the samples contain a higher number of larger particles (30–200 µm) as the curves are moved to the right side. However, the red mud, RM, contains a high number of smaller particles (0.7–10 µm). The reactivity of cement is often enhanced by optimizing its PSD. Finer particles with higher surface areas react more readily with water, increasing the rate of hydration. This is particularly evident when using finely ground supplementary cementitious materials (SCMs) such as slag, silica fume, or fly ash, which can improve reactivity by filling voids between larger particles and increasing the contact area for hydration reactions [54,56]. Optimizing the packing density through careful control of the PSD helps reduce voids within the cement matrix, which not only improves the material’s strength but also minimizes its porosity. Studies show that well-distributed particle sizes contribute to a denser, more uniform packing of the cement particles. This reduces the amount of unreacted water and enhances the chemical reactivity of the cement, leading to stronger and more durable concrete [56].

### 3.2. Chemical Characterization

Table 3 presents the results of the chemical composition of the collected secondary raw materials. Loss of ignition is considered to be low for the majority of samples, except for the red mud (RM), fly ash, and bottom ash (PFA and PBA) from the paper industry and fly ash (FA6) from the thermal power plant. The most important parameter for the valorization of these materials for the production of Al-rich types of cement is the content of Al_2_O_3_. The most abundant materials with Al_2_O_3_ are the fly ashes (FA1, FA2, FA3, and FA4) from thermal power plants, with Al_2_O_3_ contents around 23 wt.%, with only two exceptions where Al_2_O_3_ levels were low, around 5 wt.% (FA5 and FA6). A similar situation also occurs with the bottom ash (BA1) sample from the thermal power plant, which has a relatively high Al_2_O_3_ content, while the BA2 sample contains it in much lower levels. It can be speculated that the increased Al_2_O_3_ levels in the fly ashes are due to the clay admixtures in coal or waste paper. Red mud has a 16.8 wt.% of Al_2_O_3_ in this study. Waste water jet sand (WJI) and waterjet abrasive from the steel industry, and crushed brick rejects (CW1) from the brick factory have between 16 and 21 wt.% of Al_2_O_3_. Other materials have lower Al_2_O_3_ levels, around 10 wt.%, which means they were depleted even compared with the average Al_2_O_3_ content in the Earth’s upper crust.

In comparison to Ordinary Portland cement (OPC), the most used binder in the world, for the production of the low-CO_2_ non-Portland binders (belite–CSA (BCSA) binders), an approx. three times higher Al_2_O_3_ content is needed. In particular, the use of alumina-rich wastes can lower the manufacturing costs of BCSA cement mainly depending on the use of the “expensive” bauxite [57].

Considering the content of SiO_2_ in the samples, data show that all samples are depleted in this component compared to the average Earth’s upper crust. However, all kinds of ashes collected in this study have higher SiO_2_ levels compared to the red mud and slag samples (except SL5). The Fe_2_O_3_ levels vary a lot in samples from this study—from 46.68 wt.% to less than percent. The highest Fe_2_O_3_ content is in the red mud sample RM, followed by WJ1 and SL1, while the other samples contain less than a 10 wt.% of Fe_2_O_3_. However, a significant portion of ash samples were still enriched with Fe_2_O_3_ compared to the Earth’s upper crust.

The majority of samples have elevated CaO levels compared to the Earth’s crust. Considering the CaO levels in samples from this study, all slags, ashes from the paper industry, and fly ashes from thermal power plants (FA6 and FA2) can be regarded as CaO-rich materials, with a CaO content between 20 and 47 wt.%, while red mud and the rest of the ashes from the coal power plant can be regarded as materials with a lower CaO content. The studied ashes FA4, FA5, and BA2 are classified as siliceous fly ash, with less than 10% calcium oxide (CaO), whereas the calcareous ash samples (PFA, FA1, FA2, FA3, FA6, PBA, and BA1) contain more than 10% CaO [58]. According to EN 450-1 [59], the sum of SiO_2_, Al_2_O_3_, and Fe_2_O_3_ must not be less than 70%, a requirement met by all the siliceous fly ashes (FA1, FA3, FA4, and FA5) and the calcareous fly ash FA2. The sulfate contents of all the studied ashes were below the specified limit (<3%), as were the total alkali contents (Na_2_Oeq < 5%), phosphate contents (<5%), and MgO contents (<4%). The category C LOI (<5%) criterion was met by all the fly ashes except for the calcareous fly ash FA6, which also exceeded the broader limit of category C LOI < 9%. Similar conclusions can also be made for the MgO content in the waste materials from this study, with the exception that the MgO levels in many samples are lower than those from the Earth’s upper crust. According to the EN 197-1 standard [58], the content of MgO in cement shall not exceed more than 5 wt.%. All the investigated slags are characterized by a higher MgO content (in the range from 5.29 wt.% for SL5 to 11.32 wt.% for SL2), as well as WJ1. Magnesium oxide (MgO) in cementitious materials serves as an expansive agent, helping to control shrinkage, reduce cracking, and enhance durability by improving the microstructure and reducing porosity [60,61]. However, excessive MgO can cause overexpansion, cracking, and long-term reductions in compressive strength, with these effects influenced by the curing conditions and dosage levels [61,62]. The P_2_O_5_, K_2_O, SO_3_, and Na_2_O levels are generally very low, but this is not so significant for the cement industry. Alkalis (Na_2_O and K_2_O) play a significant role in the alkali–silica reaction (ASR), with the maximum alkali content in concrete often limited to ≤3.0 kg/m^3^ of Na_2_Oeq (calculated as Na_2_Oeq = Na_2_O + 0.658 K_2_O) in cement and the aggregates for exposure conditions susceptible to ASR. Some national annexes or complementary standards suggest limiting cement alkalis to ≤0.60% Na_2_Oeq by mass when reactive aggregates are present [63]. Also, EN 450-1 [59] limits the alkali content of Class F fly ash used in concrete to 1.5% Na_2_O_3_eq to control the risk of ASR, although its effectiveness also depends on factors such as ash reactivity and the type of aggregates used. In this respect, only the RM shows a higher content in comparison to the other SRMs. The highest TiO_2_ levels were measured in the red mud, while the other samples have TiO_2_ levels within the range of those in the Earth’s upper crust. Titanium dioxide (TiO_2_) enhances cementitious materials by offering photocatalytic properties, which enable self-cleaning, air-purifying, and antimicrobial features, making it valuable in eco-friendly construction. It also influences hydration by accelerating gel formation at early stages and improves microstructure, though excessive amounts may slow down the hydration process by reducing the availability of water to cement particles, potentially affecting the final setting time and overall durability. Additionally, a high TiO_2_ content may increase the risk of material degradation in environments with high UV exposure [64].

### 3.3. Trace Elements, REEs/Heavy Elements of Samples

The results of minor, trace, and rare earth elements (REEs) for the investigated samples are presented in Table 4.

### 3.4. Comparison of Different Industrial Residues Based on Minor and Trace Elements

Steel industry slags (Ss) stand alone among all the analyzed industrial residues since they contain high average levels of Cr and Mn (14,600 and 16,300 mg/kg, respectively). Cr and Mn are alloying elements that enhance the quality of steel. Minor amounts of alloying elements (Cr, Mn, etc.) can be incorporated in the slag, which originates from recycled scrap or from secondary metallurgy. In comparison to the Earth’s upper crust are steel slags enriched with Cr and Mn [65] (Figure 5). Both these elements are incorporated in EAF C slags and mixtures that contain EAF S slag (Table 4—samples SL1, SL2, and SL3). The detected levels are comparable to the other EAF slags [66,67]. High concentrations of Mn were also detected in blast furnace slag (Table 4—sample SL5). Chemical speciation and mineralogical phases that contain these elements play crucial roles in the evaluation of the reactivity and availability of elements. Mineralogical analyses of the collected samples showed that chromite and Mg–chromite are the main carriers of Cr, while Mn is probably incorporated in a solid solution of (Fe, Mg, and Mn)O. Additionally, in the case of the magnesium chromite partial replacement of Cr^3+^ by Al^3+^, Mg^2+^ by Mn^2+^ can also occur [67]. Besides Cr and Mn, slags also show considerable enrichments with Nb, Ni, V, Cu, Sb, and Zn.

Red mud (RM), as the material with the second highest enrichment with minor and trace elements, shows much lower elemental levels in comparison to the steel industry slags. The highest levels among the analyzed elements were detected for Mn, followed by Cr, Ni, V, and Zr (Figure 5). In comparison to the averages in Earth’s upper crust [65], RM contains 163-fold higher levels of Cd, 33-fold higher levels of As, 15-fold higher levels of Cr, and 10- to 8-fold higher levels of Pb, Ni, and V (Table 4). Similar trace element levels were also detected by Radusinović and Papadopoulos [68] for red mud in Podgorica, Montenegro. One of the issues of red mud application in the cement industry is the conversion of total Cr into water soluble Cr^6+^ [69].

In fly ashes from paper mills (FAPIs), Zn is the predominant and the most enriched trace element (Figure 5). Also considerably enriched in comparison to the averages in Earth’s upper crust [65] are Cd, Sb, Ag, Cu, Pb and Hg, showing 20- to 5-fold higher levels. In bottom ashes from paper mills (BAPIs), Sr, Cu, and Ba prevail, but only Cu is significantly enriched in comparison to the averages in Earth’s upper crust. Other enriched elements are Ag, Cd, and Pb (Table 5). The fly ash from paper mills generally shows higher enrichments of the listed elements than the bottom ash from paper mills. The source of the listed elements is the high-temperature combustion of hog fuel in boilers. This fuel can also be mixed with grass plants, sugar and oil crops, agricultural residues, residues from the food and paper industries, municipal green wastes, sewage and de-inking sludge, and organic wastes and residues [70], which may contribute to the enrichment of the above listed elements. In our case, the sample PFA is combustion residue of burning the mixture of de-inking fiber paper sludge, waste wood, bark, coal, and sewage sludge, while the sample PBA is the combustion residue of de-inking and sewage sludge and natural gas [71]. Higher enrichments of PFA with trace elements might be connected with the application of coal in the burning mixture.

Fly ashes from thermal power plants (FATPs) and bottom ashes from thermal power plants (BATPs) contained the largest amounts of Mn, Ba, and Sr (Figure 5), but they do not show any significant enrichment in comparison to the Earth’s upper crust. The most enriched elements in FATP are As, Cd, Cu, and Ni, while BATP shows slight enrichment with Zn. The samples BA2 and FA3 also showed high enrichments of Hg (42-fold and 16-fold, respectively), while the other samples contain Hg levels comparable to the natural background. Slight enrichments were also detected for Be, Co, Ga, Pb, Sr, Th, V, and Zn. Those enrichments are characteristic of the coal fired thermal plants [72] and reflect the composition of the coal which is used as fuel [73].

Among the other samples (O), only WJ1—waste water jet sand from a waterjet cutting machine—shows significant enrichments of Cr, Mn, Ni, and some REEs (Table 4), which are consequences of the almandine (garnet group) sand used in the process and steel particles mixed in the material. The residuals from quartz sand washing (QS1) and brick rejects (CW1) do not contain any enrichments in comparison to the Earth’s upper crust.

Minor elements like chromium (Cr), manganese (Mn), copper (Cu), and zinc (Zn) affect clinkerization by influencing the burning process and phase formation. Cr typically raises the temperature at which the liquid phase forms, potentially increasing the clinkerization temperature. In contrast, Mn and Zn lower the melting point of the raw mix, promoting early liquid phase formation and reducing the clinkerization temperature. However, excessive Zn may cause issues like kiln coating or ring formation [74,75]. Cu and Zn influence clinker phase formation by altering the crystallization of key minerals such as C_3_S and C_3_A. A high Cu content promotes the decomposition of C_3_S into C_2_S and free lime, while Zn reduces C_3_A content by forming alternative compounds. These changes impact the performance of the cement, underscoring the need for controlled levels of these minor elements in the raw materials [76,77]. Minor elements such as chromium (Cr), manganese (Mn), and zinc (Zn) influence cement hydration by altering the phase formation and hydration kinetics. High concentrations of these elements can interact with calcium silicates (C_3_S and C_2_S) and calcium hydroxide, affecting the rate of heat generation, setting time, strength development, and durability. Mn accelerates clinker hydrolysis, improving early hydration compressive strength but reducing strength after 80 days due to combined water formation, while Cr slows hydration, decreasing both the hydration degree and compressive strength after 28 and 80 days. Cu and Zn significantly retard early hydration and strength development within the first day, but exert an accelerating effect at later ages [78,79,80].

### 3.5. REE Levels in Industrial Residues

The highest REE levels (including Sc) among the analyzed industrial residues were detected for RM, reaching 1268 mg/kg (Figure 6). Light rare earth elements (LREEs; La, Ce, Pr, Nd, Pm, Sm, and Eu) present almost a 70% share of the REEs in RM. The detected levels are somehow lower in comparison to those measured in the RM in Podgorica, Montenegro, where RM contains between 1535 and 1646 mg/kg REEs [68]. The composition of red mud also depends on the composition of the processed bauxite. Karstic bauxites in the Balkan region contain between 200 and 3500 mg/kg REEs, but bauxite with a REE content below 1000 mg/kg predominates [68,81].

The REEs in RM are present in ferrotitanite, phosphate, and carbonate minerals or are adsorbed on goethite and cancrinite—the latter two are most probably the main carriers of REEs in RM [82], which also corresponds well with the mineralogical composition of our RM sample. REEs can be extracted from RM using multistage extraction with different combinations of acid leaching, acid roasting, and high temperature smelting [8]. The main objective of the process is to remove or isolate major elements such as Fe, Al, and Ca from the REEs, which can then be selectively leached. The efficiency of such multistage extractions is between 40% and 80% [11,83,84]. However, this process was tested only at a laboratory scale, and pilots in real environments have not been utilized yet due to its economic feasibility, and the environmental impact of the process is still questionable. However, RM is an appropriate target for REE extraction when methods for industrial extraction will be available.

The average levels of REEs in the other industrial residues are 161 mg/kg for slag (S), 132 mg/kg for paper industry fly ash (FAPI), 220 mg/kg for thermal power plant fly ash (FATP), 140 mg/kg for paper industry bottom ash (BAPI), 188 mg/kg for thermal power plant bottom ash (BATP), and 150 mg/kg for other (O) materials. The content of REEs in these materials does not represent any economic potential nor environmental issue since they are directly comparable to their averages in Earth’s upper crust, which is 183 mg/kg [65].

### 3.6. Presence of Organic Matter (TOC)

Another important parameter that defines the chemical composition of raw material is the presence of organic matter, expressed by the total organic content (TOC) analyzed (Table 5). It affects mainly emissions, as in unstable operating conditions the presence of TOC in clinker raw mixture can contribute to the CO_2_ emissions. Research on the specific impact of TOC on clinker has shown that elevated levels can hinder proper sintering or lead to incomplete reactions, impacting the cement’s strength and durability. Furthermore, excess organic carbon might contribute to carbon emissions during the production process, which is a significant concern for the cement industry’s environmental footprint [85]. A balance must be struck, as excessive organic matter (above a certain threshold) can lead to lower strength gain and even longer-term deterioration. Some studies suggest that organic matter content above approx. 5% in raw materials can begin to degrade cement performance by disrupting clinker chemistry, increasing energy consumption during combustion, and introducing impurities that weaken the final product while also exacerbating environmental emissions [86]. However, the specific threshold can vary depending on the type of cement and other compositional factors in the raw materials.

The results showed (Table 5) that the highest values of TOC in secondary raw materials (red mud, slag, fly and bottom ash, etc.) are between 0.02 and 3.6%. Some samples contained a higher amount of TOC, especially fly ash from the paper industry (PFA) which contains 10.07%, which is prescribed as the remains of cellulose.

### 3.7. Mineralogical Composition

The mineralogical composition of the investigated samples is presented in Table 6, Table 7, Table 8, Table 9 and Table 10. Figure 7 presents the XRD patterns of the selected Al-containing industrial residues.

As can be seen from Table 6, the sample of red mud (RM) contained hematite, cancrinite, and goethite as major mineral phases, but also ilmenite, katoite, rutile, illite, and quartz in minor quantities.

In all steel slag samples (SL1, SL2, SL3, and SL4), the main phase was belite, as shown in Table 7. Other phases that occurred in higher amounts in most samples were ferrite, calcite, mayenite, and dolomite. Namely, the other phases present in SL1 were ferrite, wuestite and magnetite. In SL3, besides belite, the other phases present were Mg–chromite, periclase, calcite, dolomite, quartz, and ferrite. In SL2 (mixture of EAF C and ladle slag), in addition to belite, the other phases were calcite, mayenite, ferrite, and dolomite. The ladle slag SL4 contained over 50 wt.% of belite, and the other phases were mayenite, γ-belite, and gehlenite. In some steel slag samples, traces (<3 wt.%) of dolomite, merwinite, galenite, hematite, chromite, periclase, mayenite, γ-belite, gehlenite, wuestite, quartz, and magnetite were also present. The blast furnace slag sample SL5 consisted only of amorphous content (100.0 wt.%).

As regarding fly ashes, shown in Table 8, in fly ash from the paper industry the most abundant phases were ferrite and belite, followed by quartz, lime, and gehlenite. The other phases were albite, calcite, mayenite, and orthoclase.

In samples of fly ash from the thermal power plant, the predominant phase in most materials was plagioclase (FA1 and FA4) or quartz (FA3, FA5, and FA6). The other phases determined were, e.g., hematite, mullite, illite/muscovite, and in small amounts, gehlenite, lime, anhydrite, feldspar, calcite, dolomite, portlandite, and periclase.

In bottom ash from the paper industry (PBA), as shown in Table 9, the main phases were calcite and belite, followed by lime, portlandite, and gehlenite. In small amounts, mayenite, talc, anhydrite, and quartz were present.

Quartz was the main phase in the bottom ash from the thermal power plant, BA2, while in BA1 the main phase was plagioclase. The dominant phases in PBA were calcite, followed by lime, portlandite, and gehlenite. Other minor phases present in some samples of bottom ash in a quantity of less than 3 wt.% were belite, lime, K-feldspar, hematite, anhydrite, talc, quartz, mayenite, and gehlenite.

Other collected industrial residues have different compositions, as shown in Table 10. In a sample of by-product of washing quartz sand, QS1, the main phase was quartz, followed by small amounts of illite/muscovite and kaolinite. In the sample WJ1, the main phase was almandine, with small amounts of quartz and ilmenite. The sample CW1 contained quartz, plagioclase, and illite/muscovite, and small amounts of hematite and calcite.

Gamma belite, a form of calcium silicate, has a lower reactivity than alite, which can slow hydration and hinder early strength development when used as an additive in cementitious materials. It is more effective as a raw material in clinker production, where its presence can be better controlled to avoid negative impacts on hydration and strength development [21,87]. Periclase, composed of magnesium oxide, can disrupt the clinkerization process and slow hydration rates when used as an additive in cement, negatively affecting early strength and durability. Its instability under certain conditions can also lead to expansion issues, compromising the long-term integrity of cement-based materials [88]. Therefore, while periclase might be useful in specific contexts, its presence in cement needs to be carefully controlled to avoid these potential problems.

Free lime in ashes, such as fly ash, can contribute to additional cementitious reactions, improving strength and durability. However, too much can reduce strength and cause issues like shrinkage and increased water absorption. For optimal performance, free lime content in ash should generally be kept below 10%, as higher levels can lead to expansion and durability problems [89,90].

### 3.8. The Results of Radiological Characterization

Table 11 shows the results of the radiological characterization of the secondary raw materials. Except for ^137^Cs, which was detected in a few samples, all detected radionuclides have a natural origin. The measured concentrations of ^137^Cs are low and are a consequence of the Chernobyl accident in 1986. The measured values of ^226^Ra, ^232^Th, and 40K are characteristic for the appropriate type of samples and correspond to the values measured in other tests [36,91,92,93].

Based on the results shown in Table 11, the external hazard index (*H_ext_*), internal hazard index *H_int_*, and the activity concentration index (*ACI*) were calculated, and their values are presented in Table 12.

It must be noted that the permitted limits of radioactive elements depend on the purpose of the material and are limited based on the estimated doses. To assess the potential health impact on the public due to exposure to the tested samples for which the *ACI* is greater than 1, the annual effective dose (*E*) of the total external absorbed gamma dose in air at a height of 1 m above ground level for outdoor and indoor cases (*p* = 0.2, *E_terr,20%_* and *p* = 0.8, *E_terr,80%_*) and for four different cases of standard rooms (all walls, floor and ceiling, only floor and walls, and only floor made of them, as well as only superficial materials) was calculated and is shown in Table 13.

It can be concluded from a radiological point that with the exception of RM (red mud), FA1 (fly ash from the thermal power plant), and BA1 (bottom ash from the thermal power plant), all the other tested samples met both of the strictest criteria: *ACI* < 1 and *Hint* < 1. Accordingly, the criteria of *R_aeq_* < 370, *H_ex_t* < 1, and *I_α_* < 1 are also met for them, as well as the effective dose for a standard model room being less than 1 mSv/y in the case that all walls, the floor, and the ceiling are built with them. From the point of view of radioactivity, they can be freely used as building materials. For samples with *ACI* > 1, the effective dose was calculated for the external exposure of gamma rays in air at the height of 1 m above ground level for outdoor and indoor cases (*E_terr,20%_*, and *E_terr,80%_*) and four different cases of standard rooms (*E_all_*, *E_fw_*, *E_f_*, and *E_sup_*). Considering that all the calculated doses *E_terr,20%_* are less than 1, all the tested materials are safe from the aspect of external terrestrial exposure, with the limitation of the duration of exposure being less than 20% of the hours per year. Also, all the tested materials can be used to make only the floor, and as a surface material (*E_f_* and *E_sup_* are less than 1).

If the hypothetical situation occurs where the annual effective dose for all situations is greater than one, whether the material can be disposed of in the environment is examined (the criteria are defined by competent institutions in the specific country). If it turns out that the material is not safe for disposal, it is reported to a competent institution (usually the radiological inspection), which passes measures on further action.

## 4. Conclusions

These characterized (from their physical, chemical, mineralogical, and radiological aspects) 19 samples of industrial Al-containing residues from the ESEE region present a base for their advantages as alternative raw materials that can influence improvement of the properties/add value on the final products as well as of the cost-effectiveness and environmental stability of cements. This paper’s results tackle two aspects of secondary raw materials, the environmental (heavy metals) and radiological aspects, which are mostly separately considered in practice.

The characterized Al-containing residues from the ESEE region showed the potential to be used in construction and beyond. Up to now, globally, fly ash has been mostly used in cement and concrete production, but also the other characterized Al-containing residues could be successfully utilized either in the construction and building industries or in wider applications for traditional building products, like bricks, glass–ceramics, tiles, etc., or for products supporting the green transition like geopolymers, aerogels, zeolites, REEs, etc. The proportions of each Al-containing residue should be calculated based on the needs of the final material/product to achieve the necessary goal. In this respect, pre-treatment of such Al-containing residues could have a beneficial effect.

Future investigations need to be performed for creative applications of the characterized Al-containing residues, as they seem to be particularly promising in relation to environmental benefits, such as lowering CO_2_ emissions, controlling greenhouse emissions, and reducing environmental contamination.

## Figures and Tables

**Figure 1 materials-17-06245-f001:**
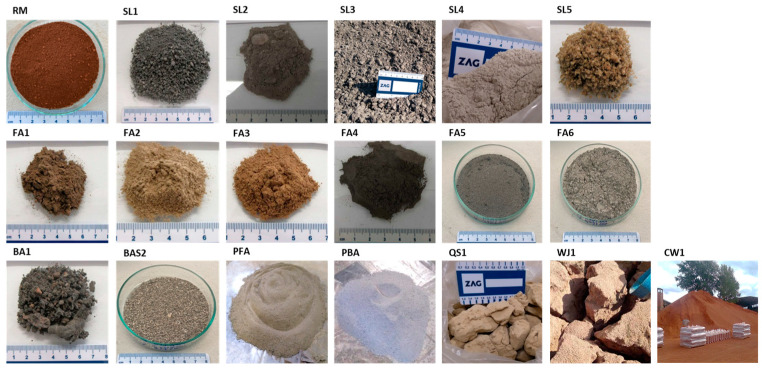
Samples of collected secondary raw materials.

**Figure 2 materials-17-06245-f002:**
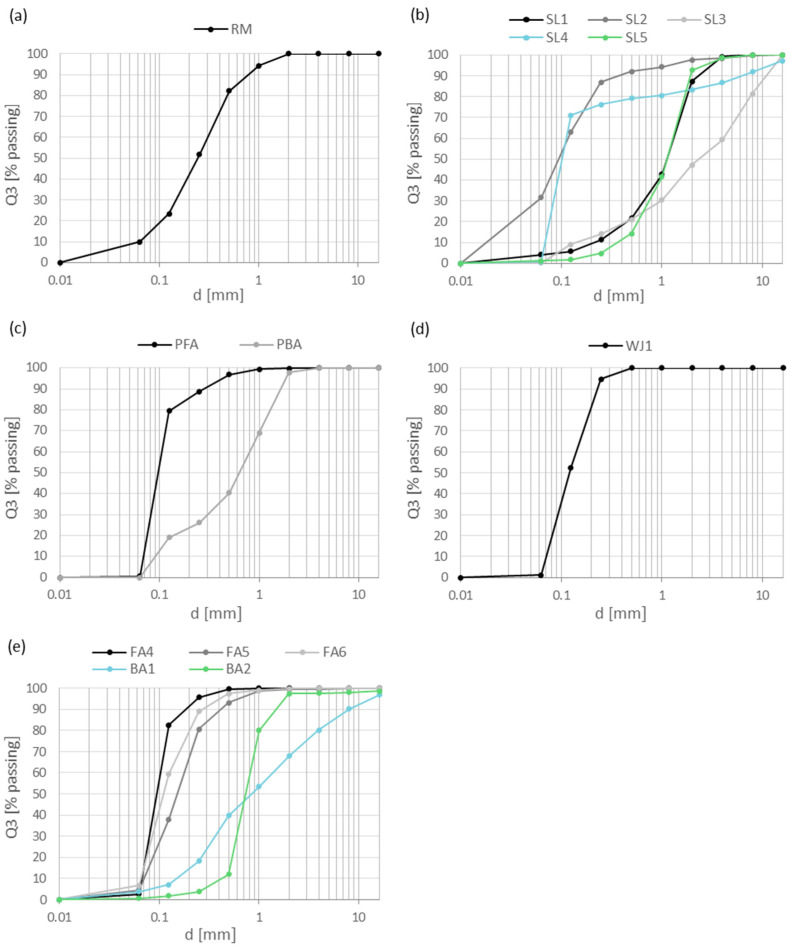
Sieve analysis for Al-containing industrial residues: (**a**) red mud, (**b**) steel slags, (**c**) ashes from the paper industry, (**d**) waste water jet sand, and (**e**) ashes from thermal power plants.

**Figure 3 materials-17-06245-f003:**
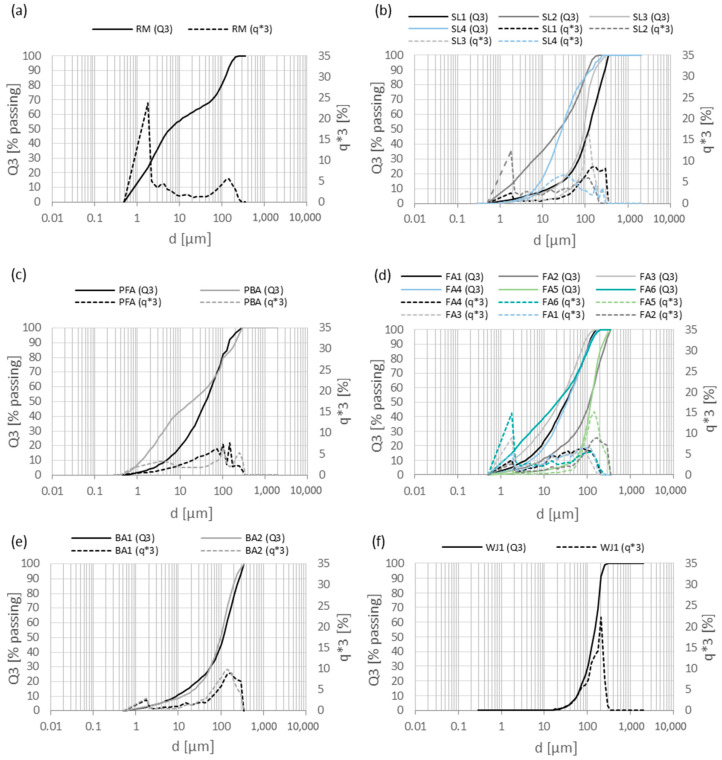
PSDs of (**a**) red mud from alumina production, (**b**) slags from still production, (**c**) fly and bottom ashes from the paper industry, (**d**) fly ashes from thermal power plants, (**e**) bottom ashes from thermal power plants, and (**f**) waste water jet sand.

**Figure 4 materials-17-06245-f004:**
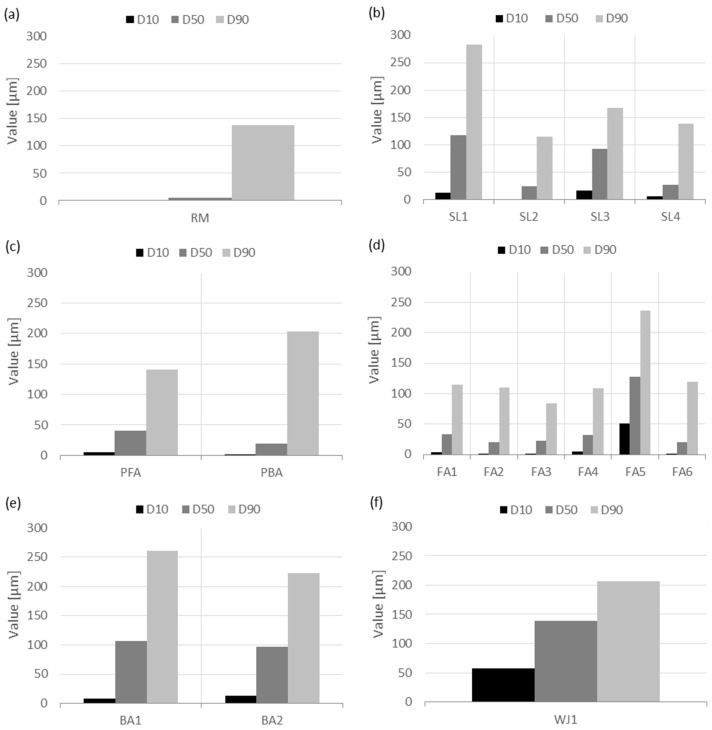
PSDs (D10, D50, and D90) of (**a**) red mud from alumina production, (**b**) slags from still production, (**c**) fly and bottom ashes from the paper industry, (**d**) fly ashes from thermal power plants, (**e**) bottom ashes from thermal power plants, and (**f**) waste water jet sand.

**Figure 5 materials-17-06245-f005:**
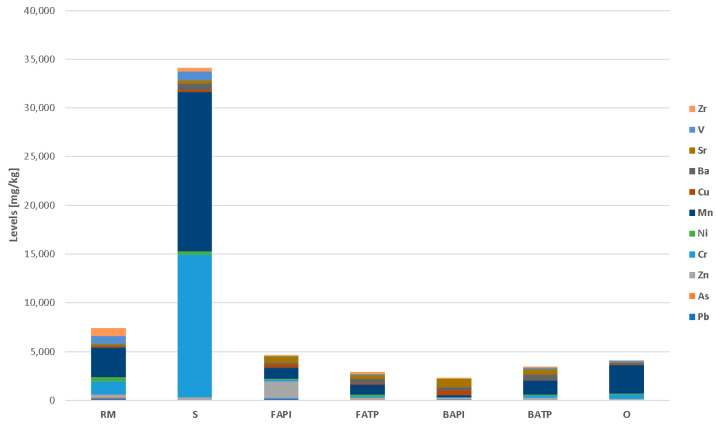
Average levels of selected minor and trace elements in the analyzed industrial wastes. Units in mg/kg.

**Figure 6 materials-17-06245-f006:**
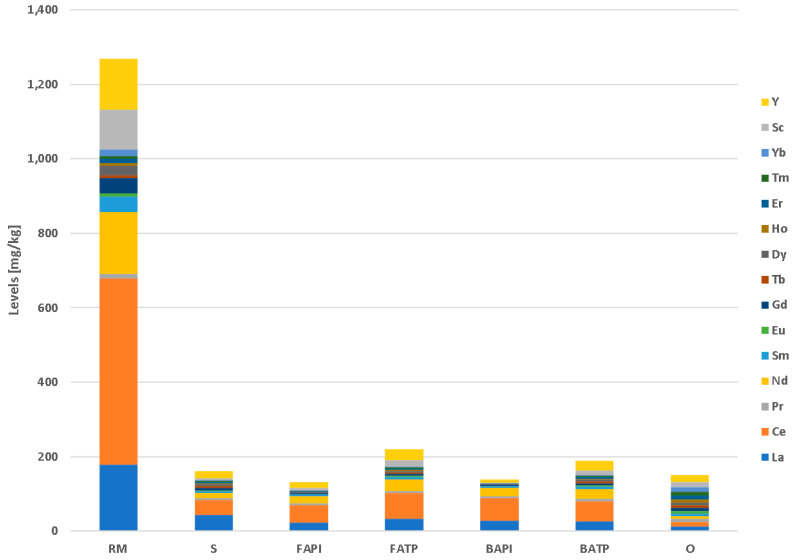
Average levels of REEs in the analyzed industrial wastes. Units in mg/kg.

**Figure 7 materials-17-06245-f007:**
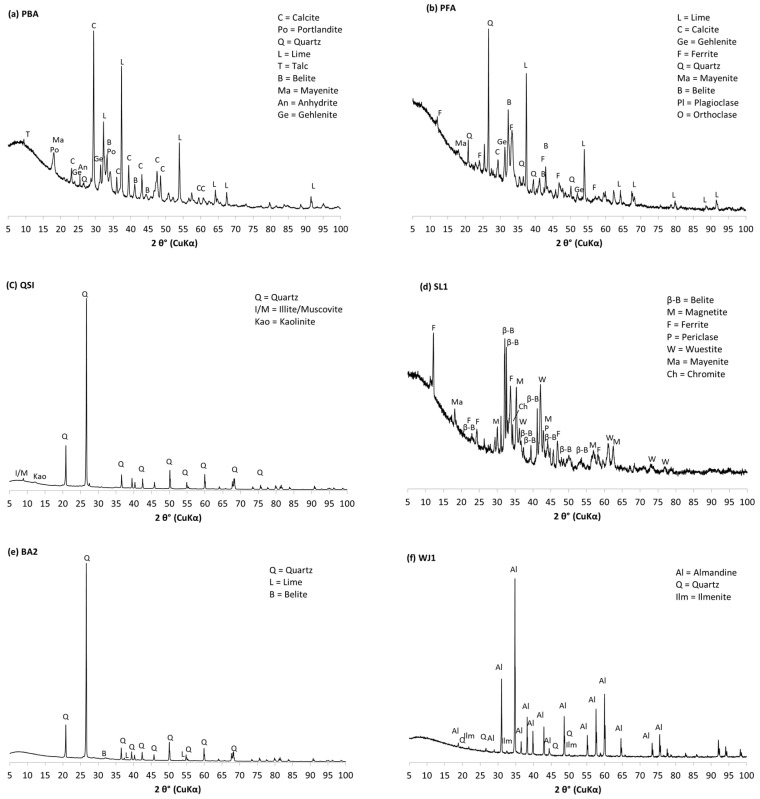
XRD patterns for selected samples. (**a**) Bottom ash from the paper industry (PBA), (**b**) coal fly ash (PFA), (**c**) by-product of washing quartz sand (QSI), (**d**) EAF C steel slag (SL1), (**e**) coal bottom ash (BA2), and (**f**) waste water jet sand (WJ1).

**Table 1 materials-17-06245-t001:** Coding of the samples.

Sample ID	Date of Sampling	Origin	Type of Sample	Company/Location	Country
RM	October 2019	Alumina production	Red mud	Dobro Selo, Mostar	Bosnia and Hercegovina
SL1	October 2019	Steel industry	EAF C steel slag	SIJ Acroni, Jesenice	Slovenia
SL2	February 2020		Slag mineral residue (mixture of EAF S slag and ladle slag)	SIJ Acroni, Jesenice	Slovenia
SL3	October 2020		EAF C steel slag	SIJ Metal, Ravne	Slovenia
SL4	October 2020		Ladle slag	SIJ Metal, Ravne	Slovenia
SL5	October 2019		Blast furnace slag	Arcelor Mittal, Zenica	Bosnia and Hercegovina
FA1	November 2019	Thermal power plants	fly ash	Šoštanj power plant, Šoštanj	Slovenia
FA2	October 2019		Fly ash	Power plant Kakanj, Kakanj	Bosnia and Hercegovina
FA3	October 2019		Fly ash	Power plant Stanari, Stanari	Bosnia and Hercegovina
FA4	June 2020		Fly ash	TITAN Usje, Skopje	North Macedonia
FA5	September 2020		Fly ash	Anonymous	Hungary
FA6	September 2020		Fly ash		Hungary
BA1	November 2019		Bottom ash	Šoštanj power plant, Šoštanj	Slovenia
BA2	September 2020		Bottom ash	Anonymous	Hungary
PFA	September 2020	Paper mills	Fly ash	Vipap Videm, Krško	Slovenia
PBA	September 2020		Bottom ash	Vipap Videm, Krško	Slovenia
QS1	September 2020	Mining company for the production and processing of silica sands and the production of auxiliary casting material	By-product of quartz sand washing	Termit, Moravče	Slovenia
WJ1	September 2020	Steel industry	Waste water jet sand	SIJ Acroni, Jesenice	Slovenia
CW1	September 2020	Brick factory	Crushed brick rejects	Goriške opekarne, Renče	Slovenia

**Table 2 materials-17-06245-t002:** Moisture content, particle density, bulk density, and BET specific surface area (SSA) of investigated samples. n.a.—not analyzed.

Type of Sample	Sample ID	Moisture Content [%]	Particle Density [g/cm^3^]	Bulk Density [kg/m^3^]	BET SSA [m^2^/g]
Red mud (RM)	RM	0.21	3.58	1100	13.05
Slag (S)	SL1	0.3	2.66	1810	1.56
	SL2	17.5	3.25	1281	3.51
	SL3	2.83	3.39	1835	2.98
	SL4	0.00	3.04	1138	0.61
	SL5	0.08	2.62	1032	0.58
Fly ash— paper industry (FAPI)	PFA	0.17	3.05	728	51.98
Fly ash— thermal power plant (FATP)	FA1	0.01	2.34	1035	2.41
	FA2	0.05	2.47	1100	0.97
	FA3	1.89	2.40	478	6.11
	FA4	0.03	1.91	612	7.26
	FA5	0.00	2.62	1278	1.80
	FA6	0.00	2.58	620	14.33
Bottom ash— paper industry (BAPI)	PBA	0.15	2.76	529	5.23
Bottom ash— thermal power plant (BATP)	BA1	0.12	2.52	698	10.37
	BA2	0.003	2.64	1261	0.28
Other (O)	QS1	20.8	2.66	n.a.	n.a.
	WJ1	5.65	4.14	2206	0.76
	CW1	8.14	2.76	n.a.	n.a.

**Table 3 materials-17-06245-t003:** Chemical composition of samples.

Type of Sample	Sample ID	LOI 950 °C	Al_2_O_3_	SiO_2_	Fe_2_O_3_	CaO	MgO	P_2_O_5_	K_2_O	SO_3_	Na_2_O	TiO_2_
RM	RM	14.05	16.21	3.28	46.68	12.91	0.45	0.10	0.32	0.06	2.56	3.46
S	SL1	0.48	8.88	9.82	30.96	32.68	10.04	0.34	0.01	0.24	0.06	0.36
	SL2	5.12	9.70	16.84	10.33	39.12	11.32	0.12	0.03	0.35	0.13	0.68
	SL3	5.77	11.38	15.91	20.82	21.61	10.15	0.13	0.15	0.23	0.26	0.21
	SL4	0.00	19.51	17.16	1.03	52.82	7.89	0.01	0.01	0.16	0.09	0.11
	SL5	0.00	8.68	40.75	1.12	39.75	5.29	0.03	0.74	0.22	0.26	0.26
FAPI	PFA	13.86	9.55	22.29	8.86	35.86	5.3	0.29	0.84	1.99	0.46	0.96
FATP	FA1	0.57	23.23	44.60	9.86	13.38	2.68	0.45	1.8	1.46	0.92	0.83
	FA2	0.26	21.34	43.76	7.49	20.01	2.3	0.32	1.42	1.69	0.36	0.65
	FA3	3.33	23.74	48.49	7.42	11.52	3.12	0.06	3.2	1.39	0.11	1.21
	FA4	1.48	23.07	53.30	8.23	4.89	2.14	0.20	2.69	0.39	1.11	0.89
	FA5	1.89	5.78	72.30	2.20	8.91	1.56	0.61	3.76	0.38	0.82	0.56
	FA6	12.32	5.69	38.34	1.64	26.37	3.65	1.95	7.93	2.11	0.64	0.35
BAPI	PBA	15.41	8.42	13.89	0.49	58.89	1.94	0.28	0.31	0.33	0.30	0.29
BATP	BA1	2.93	22.14	41.44	10.38	16.01	2.86	0.49	1.40	0.63	0.70	0.86
	BA2	0.59	1.84	79.74	0.55	9.61	1.39	0.74	3.44	0.14	0.14	0.09
O	QS1	2.56	6.30	86.61	1.12	1.51	0.36	0.02	0.88	0.09	0.09	0.24
	WJ1	0.00	21.03	35.12	35.09	2.58	5.84	0.04	0.01	0.05	0.04	1.99
	CW1	1.49	15.96	63.29	7.52	5.17	1.79	0.11	2.09	0.23	0.87	0.84

**Table 4 materials-17-06245-t004:** Results of minor, trace, and rare earth elements in the samples of red mud (RM), slag (S), fly ash from the paper industry (FAPI), fly ash from thermal power plants (FATP), bottom ash from the paper industry (BAPI), bottom ash from thermal power plants (BATP), and other samples (O). The last column B (background) indicates the averages in Earth’s upper crust [65].

Sample ID		RM	S					FAPI	FATP						BAPI	BATP		O			B
		RM	SL1	SL2	SL3	SL4	SL5	PFA	FA1	FA2	FA3	FA4	FA5	FA6	PBA	BA1	BA2	QS1	WJ1	CW1	
Minor and trace elements (mg/kg) *	**Pb**	184	˂50	82	92.7	11.2	˂20	196.5	55	79	˂20	30.9	<30	22.7	109.1	˂20	<30	13.8	9.2	33.8	17
	**As**	157	˂25	˂20 ^b^	3.3	<0.2	˂10	17	54	108	˂10	67.5	<30	<20	2.4	˂10	<30	4.1	2.3	13.4	4.8
	**Zn**	258	88	460	686.4	52.9	33	1733.6	157	215	50	103	112	338	174.6	83	130	37.1	241.8	109.5	67
	**Co**	60.3	4	22	16	9.3	˂1	48.5	16	25	48	22.1	<10 ^e^	4	4.1	16	<10	3.8	42.3	27.7	17.3
	**Cd**	14.7	˂5	˂2	1.5	<0.02	˂2	1.8	˂2	˂2	˂2	<2	<3	3.3	0.5	˂2	<3	0.1	0.5	0.2	0.09
	**Cr**	1360	16,860	20,755	>10,000	914	83	163	89	311	531	97.2	53	49	33	325	215	51	1082	147	92
	**Ni**	435	324	1050	259.5	42.1	19	97.7	40	303	534	60.3	19.3	31	13	93	76.5	15.3	253.5	113.4	47
	**Sb**	˂50	˂50	˂40 ^b^	3.1	2.7	˂10	6.2	˂10	˂10	˂10	<10	<15	<10	2.7	˂10	<15	0.6	0.4	0.8	0.4
	**Mn**	3020	31,053	15,206	>10,000	802	24,049	1155	1717	416	990	721	805	1632	195	2634	500	81	7005	1683	774.5
	**Cu**	101	307	253	232.5	95.9	54	335.9	71	123	161	81.6	64	112	482	77	54.6	9.1	80.6	61	28
	**Ag ^#^**	˂25	˂25	˂10	- ^c^	96	˂10	680	˂10	˂10	˂10	<10	<15	<10	820	˂10	<15	46	<20	81	53
	**Ba**	104	1001	351	188	291	1411	161	513	469	420	674	267	417	360	510	223	195	11	312	628
	**Sr**	180	316	253	292	460	429	624	572	350	256	285	230	596	820	650	247	23	10	138	320
	**Ga**	60.9	˂50 ^a^	˂15	6.8	0.5	˂15	11	45	53	54	34.7	<15	<10	10.5	39	<15	5.5	10.7	19.5	17.5
	**Nb**	77.9	84	346	204.6	5	˂20	7.5	˂20	˂20	˂20	18.8	<15	<10	5.6	˂20	<15	3.4	38	11.3	12
	**Ta**	˂50	˂50	˂20	2.6	0.2	˂20	0.6	˂40	˂20	˂20	<20	<30	<20	0.5	˂20	<30	0.2	2.6	0.7	0.9
	**U**	˂75	˂125	˂20	3.3	4.9	˂20 ^d^	2.3	˂50	˂50	˂50	<30	<45	<30	2.9	˂50	<45	2	1.5	1.1	2.7
	**V**	812	525	404	3265	312	11	65	186	197	254	132	30.8	30.9	15	201	12.6	26	227	148	97
	**Zr**	779	249	536	618.6	412.7	100	97	152	158	310	159	248	120	109.9	147	43.1	16.1	39.2	34.9	193
	**Hg**	0.098	0.003	0.024	0.002	0.002	0.005	0.258	0.02	0.116	0.824	0.047	0.000	0.000	0.000	0,002	0.015	0.014	0.000	0.001	0.05
REEs ** (mg/kg)	**Ce**	499	˂30 ^a^	84	20.7	18.7	61	47	94	60	91	112	31	29	59.8	70	7.7	27.1	36.9	56.3	63
	**Dy**	27.1	˂10	˂5	1.1	1.2	15	3.1	˂5	˂5	6.5	6.44	<7.5	<5	1.7	6.2	<7.5	1.5	32.3	2.8	3.9
	**Er**	˂25	˂20	˂5	0.8	0.8	˂10	1.4	˂10	˂10	˂10	<10	<7.5	<5	0.8	˂10	<7.5	0.6	27.3	1.2	2.3
	**Eu**	8.31	˂5	˂2	0.2	0.2	˂3	0.8	˂3	˂3	3.3	<3	<5	<3	0.7	˂3	<5	0.5	0.2	0.8	1
	**Gd**	41.5	˂25	˂10	1.2	1.5	17	3.1	˂10	˂10	˂10	<10	<15	<10	2.5	˂10	<15	2	13.2	4.1	4
	**Ho**	˂12.5	˂12.5	˂5	0.2	0.3	˂5	0.5	˂5	˂5	˂5	<5	<7.5	<5	0.3	˂5	<7.5	0.2	7.8	0.5	0.83
	**La**	179	15	143	13.5	14.3	28	23.2	37	33	46	55.3	14	13.9	28.8	35	<5	14.3	16.1	23.5	31
	**Nd**	166	16	˂10	8	7.6	32	19	33	34	42	44.5	12.6	12.8	22	36	<10	12.4	17.2	24.3	27
	**Pr**	˂25	˂25	˂10	2.1	2.3	˂10	5.2	˂10	˂10	˂10	<10	<15	<10	5.9	˂10	<15	3.1	4.3	5.9	7.1
	**Sm**	42.6	˂25	˂10	1.6	1.3	12	4.1	10	11	13	10.4	<7.5	<5	4.4	12	<7.5	2.9	5.2	4.6	4.7
	**Sc**	108	˂2.5	1.9	2.9	2.6	16	6.7	21	20	34	17.1	5.6	4.1	2.5	21	<2	2.9	66.3	13.8	14
	**Tb**	˂12.55	˂25	˂10	0.1	<0.1	˂10	0.4	˂10	˂10	˂10	<5	<7.5	<5	0.3	˂10	<7.5	0.1	3.6	0.4	0.7
	**Tm**	˂12.5	˂12.5	˂5	0.1	0.1	˂5	0.2	˂5	˂5	˂5	<5	<7.5	<5	0.1	˂5	<7.5	<0.1	4.1	0.2	0.3
	**Yb**	16.7	˂5	˂0.5	0.9	0.8	6.2	1.4	˂5	˂5	˂5	4.16	<2	1.1	0.8	˂5	<2	0.7	26.7	1.3	1.96
	**Y**	136	2.9	5.4	8	8.9	69	16.3	32	32	48	36.4	15.3	12.2	8.9	34	4.5	6.2	182.1	12.2	21
Other elements (mg/kg)	**Li**	-	-	-	15.9	8.7	-	22.4	-	-		-	-	-	11.4	-	-	15.8	6	66.1	24
	**Be**	9.11	˂5	2.3	<1	<1	6.4	2	2.7	2.3	2.4	5.43	<3	<2	<1	2.70	<3	<1	<1	2	2.1

Notes: ^#^ Ag expressed in µg/kg; * selenium and thallium failed in the recovery test, so these values are not given; ** lutetium was used as an internal standard, so it was not measured in samples. Sample specific comments: ^a^ SL1: The cerium emission lines were strongly disturbed by the matrix, so higher LOQ values had to be used. Similarly, the high chromium content disturbed the Ga 294.364 nm line, so a higher LOQ value had to be used. In the case of tantalum at the first parallel measurement, the more sensitive emission line (Ta 240.063 nm) was saturated by Fe 239.924 nm. Later on, this effect was eliminated. ^b^ SL2: The usual and more sensitive As 188.979 line was disturbed by the matrix, so that a higher LOQ value had to be used. Both of the utilized antimony lines (Sb 206.836 and Sb 217.582 nm) were disturbed by the matrix, so higher LOQ values had to be used. These LOQ values are lower compared to other samples. This is the result of the method development between the different sets of samples. ^c^ SL3: The analytical result of Ag could not be provided due to unusually high levels of interference from other elements. ^d^ SL5: The uranium LOQ value is lower compared to the other samples. This is the result of the method development between the different sets of samples. ^e^ FA5: The cobalt spectrum was strongly disturbed, so a higher LOQ had to be used.

**Table 5 materials-17-06245-t005:** Results of TOC measurements.

Type of Sample	Sample ID	TOC (%)
RM	RM	0.16
S	SL1	0.05
	SL2	0.28
	SL3	0.22
	SL4	0.02
	SL5	0.04
FAPI	PFA	10.07
FATP	FA1	0.16
	FA2	0.35
	FA3	0.30
	FA4	3.61
	FA5	0.29
	FA6	1.3
BAPI	PBA	0.45
BATP	BA1	1.32
	BA2	0.05
O	QS1	0.11
	WJ1	0.02
	CW1	0.03

**Table 6 materials-17-06245-t006:** Mineralogical composition of red mud.

Sample ID	Phases, wt.%											
	A	C	H	Ca	Ka	Il	Go	Bo	I	Q	R	Sum
RM	14.2	27.1	27.0	16.4	2.6	2.1	13.1	5.5	1.5	1.1	0.2	100

A—amorphous; C—calcite; H—hematite; Ca—cancrinite; Ka—katoite; Il—ilmenite; Go—goethite; Bo—boehmite; I—illite; Q—quartz; R—rutile.

**Table 7 materials-17-06245-t007:** Mineralogical composition of slags.

Sample ID	Phases, wt.%																
	A	β-B	γ-B	F	C	W	M	P	Ma	Ch	M-C	D	Q	H	Ga	Me	SUM
SL1	4.7	38.7	0.4	25.9	/	14.1	11.1	2.2	1.9	1.0	/	/	/	/	/	/	100.0
SL2	35.8	24.0	/	9.6	11.2	0.4	/	/	10.4	0.6	/	6.2	0.8	/	/	/	100.0
SL3	*	29.9	3.5	5.8	11.2	/	/	12.8	/	/	14.0	11.2	10.2	2.3	1.1	/	100.0
SL4	*	53.0	11.7	/	/	/	0.5	0.3	25.5	/	/	2.4	/	0.3	4.0	2.3	100.0
SL5	100	/	/	/	/	/	/	/	/	/	/	/	/	0		/	100.0

A—amorphous; β-B—β-belite; γ-B—γ-belite; F—ferrite; C—calcite; W—wuestite; M—magnetite; P—periclase; Ma—mayenite; Ch—chromite; M-C—Mg–chromite; D—dolomite; Q—quartz; H—hematite; Ga—galenite; Me—merwinite; *—amorphous phase was not considered.

**Table 8 materials-17-06245-t008:** Mineralogical composition of collected fly ash samples.

Sample ID		Phases, wt.%																		
	A	Ge	F	Pl	C	Q	O	Ma	L	Mu	H	K	An	Mf	Et	I/M	Do	Po	P	SUM
PFA	43.8	6.0	14.8	3.9	1.9	9.4	0.4	0.8	8.3	/	/	/	/	/	/	/	/	/	/	100.0
FA1	79.5	0.6	/	7.9	/	4.1	/	/	0.2	6.7	0.6	0.4	5.5	/	/	/	/	/	/	100.0
FA2	81.9	/	/	/	/	5.0	/	/	2.6	/	4.5	/	/	0.5	/	/	/	/	/	100.0
FA3	80.4	/	/	2.8	/	10.4	/	/	/	5.3	0.2	/	/	/	0.9	/	/	/	/	100.0
FA4	64.9	/	/	18.0	/	9.0	/	/	/	/	0.6	/	/	/	/	7.3	0.2	/	/	100.0
FA5	35.4	/	/	11.1	0.8	48.1	/	/	0.3	/	/	4.3	/	/	/	/	/	/	/	100.0
FA6	51.7	/	/	5.4	18.0	19.0	/	/	1.2	/	/	1.0	/	/	/	/	1.1	1.4	1.2	100.0

A—amorphous; Ge—gehlenite; F—ferrite; Pl—plagioclase; C—calcite; Q—quartz; O—orthoclase; Ma—mayenite; L—lime; Mu—mullite; H—hematite; K—K-feldspar; An—anhydrite; Mf—magnesioferrite; Et—ettringite; I/M—illite/muscovite; Do—dolomite; Po—portlandite; P—periclase.

**Table 9 materials-17-06245-t009:** Mineralogical composition of collected bottom ash samples.

Sample ID		Phases, wt.%													
	A	C	Po	Q	L	T	B	My	An	Ge	Pl	H	K	B	SUM
PBA	26.5	23.0	9.8	0.3	12.0	0.7	20.1	1.6	0.4	5.6	/	/	/	/	100.0
BA1	65.1	/	/	2.8	/	/	/	1.7	/	1.4	27.9	0.6	0.5	/	100.0
BA2	42.4	/	/	56.4	0.5	/	/	/	/	/	/	/	/	0.7	100.0

A—amorphous; C—calcite; Po—portlandite; Q—quartz; L—lime; T—talc; B—belite; My—mayenite; An—anhydrite; Ge—gehlenite; Pl—plagioclase; H—hematite; K—K-feldspar; B—belite.

**Table 10 materials-17-06245-t010:** Mineralogical composition of other collected industrial residues.

Sample ID	Phases, wt.%									
	A	Q	I/M	Kao	Al	Ilm	Pl	H	C	SUM
QS1	24.4	72.2	1.9	1.5	/	/	/	/	/	100.0
WJ1	23.2	1.0	/	/	85.0	0.8	/	/	/	100.0
CW1	51.5	31.3	7.4				8.1	1.1	0.6	100.0

A—amorphous; Q—quartz; I/M—illite/muscovite; Kao—kaolinite; Al—almandine; Ilm—ilmenite; Pl—plagioclase; H—hematite; C—calcite.

**Table 11 materials-17-06245-t011:** Radiological characteristics of the investigated samples.

Type of Sample	Sample ID	^210^Pb	^226^Ra	^232^Th	^40^K	^137^Cs	^238^U	^235^U
RM	RM	107 ± 11	176 ± 7	397 ± 25	32 ± 6	<0.1	164 ± 15	10.3 ± 1.5
S	SL1	<2	19.2 ± 1.3	3.3 ± 0.5	<1	<0.04	14.1 ± 1.7	1.00 ± 0.08
	SL2	5 ± 2	22.6 ± 1.1	6.6 ± 0.8	11.7 ± 1.7	0.6 ± 0.1	23 ± 4	1.7 ± 0.3
	SL3	6.8 ± 2.1	29 ± 2	14.7 ± 12	36.7 ± 3.2	<0.03	28.6 ± 3.0	1.8 ± 0.2
	SL4	<0.7	34.9 ± 1.5	8.2 ± 0.6	<0.7	<0.02	44.5 ± 5.6	2.5 ± 0.3
	SL5	<13	124 ± 7	24 ± 4	160 ± 20	<0.6	88 ± 13	7.0 ± 0.7
FAPI	PFA	42.4 ± 4.8	42 ± 2	32 ± 3	242 ± 17	9.9 ± 0.7	39.2 ± 5.3	1.9 ± 0.2
FATP	FA1	362 ± 24	406 ± 16	57.6 ± 4.6	562 ± 36	<0.08	364 ± 45	24 ± 3
	FA2	55.6 ± 6	24 ± 2	19 ± 3	170 ± 20	<0.2	33 ± 5	2.2 ± 0.3
	FA3	17 ± 4	25 ± 2	23 ± 3	90 ± 10	<0.2	20 ± 4	1.5 ± 0.2
	FA4	93 ± 8	99 ± 4	78 ± 5	670 ± 40	˂0.09	114 ± 10	5.4 ± 0.4
	FA5	96 ± 7	30.0 ± 1.6	22.5 ± 2.2	1260 ± 77	8.9 ± 0.7	31.5 ± 5.4	2.0 ± 0.2
	FA6	271 ± 18	42 ± 2	23 ± 3	1810 ± 110	16.3 ± 1.2	33.6 ± 6.7	1.5 ± 0.2
BAPI	BAS2	23.5 ± 3.7	33 ± 2	42 ± 3	90 ± 8	1.2 ± 0.3	34.7 ± 4.6	1.8 ± 0.2
BATP	BA1	736 ± 87	420 ± 24	56 ± 4	469 ± 31	<0.1	408 ± 27	23 ± 1
	BA2	13 ± 3	20 ± 1	10.9 ± 1.4	1030 ± 64	1.2 ± 0.2	6.8 ± 2.7	0.30 ± 0.05
O	QS1	45.5 ± 5.2	24 ± 1	13 ± 1	281 ± 18	<0.04	38.9 ± 4.7	1.8 ± 0.2
	WJ1	14.2 ± 4.3	29.6 ± 2.0	103 ± 7	8.2 ± 1.8	<0.03	26.2 ± 3.9	1.6 ± 0.2
	CW1	45.5 ± 6.9	35 ± 1.6	48 ± 3	653 ± 41	<0.04	44 ± 6	2.4 ± 0.2

**Table 12 materials-17-06245-t012:** Screening indices for an assessment of the radiological load by the investigated samples.

Type of Sample	Sample ID	*H_ext_*	*H_int_*	*ACI*
RM	RM	2.02	2.49	2.58
S	SL1	0.06	0.12	0.08
	SL2	0.09	0.15	0.11
	SL3	0.14	0.22	0.18
	SL4	0.13	0.22	0.16
	SL5	0.47	0.80	0.59
FAPI	PFA	0.29	0.41	0.38
FATP	FA1	1.45	2.55	1.83
	FA2	0.18	0.24	0.23
	FA3	0.18	0.25	0.23
	FA4	0.73	1.00	0.94
	FA5	0.47	0.55	0.63
	FA6	0.64	0.75	0.86
BAPI	PBA	0.27	0.36	0.35
BATP	BA1	1.46	2.60	1.84
	BA2	0.34	0.40	0.46
O	QS1	0.18	0.25	0.24
	WJ1	0.48	0.56	0.62
	CW1	0.44	0.53	0.57

**Table 13 materials-17-06245-t013:** Estimation of the annual effective dose for different uses of the investigated samples.

Type of Sample	Sample ID	*E_terr,80%_* (mSv)	*E_terr,20%_* (mSv)	*E_all_* (mSv)	*E_fw_* (mSv)	*E_f_* (mSv)	*E_sup_* (mSv)
RM	RM	>1	<1	>1	>1	<1	<1
FATP	FA1						
BATP	BA1						
Other	/	<1, due to ACI < 1					

## Data Availability

The original contributions presented in the study are included in the article, further inquiries can be directed to the corresponding author.

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
