# Peer review of "Characterization of Al-Containing Industrial Residues in the ESEE Region Supporting Circular Economy and the EU Green Deal"

_materials, 2024, doi:10.3390/ma17246245_

Round 1
Reviewer 1 Report
Comments and Suggestions for Authors
Summary of the Manuscript
This manuscript explores the characterization of Al-containing industrial residues in the East South East Europe (ESEE) region, with a focus on supporting the circular economy and the EU Green Deal. The study investigates the potential use of industrial residues, such as red mud, fly ash, and slag, as secondary raw materials in construction and other sectors, examining their physical, chemical, mineralogical, and radiological characteristics. This characterization aims to lay a foundation for their future utilization as sustainable materials aligned with circular economy principles.
Major Issues
- Methodology Clarity: The study involves a detailed analysis of various physical and chemical characteristics. However, the methodology could benefit from clearer descriptions, especially concerning sample preparation techniques, controls, and the consistency across samples from different industries. For instance, the paper mentions “drying at 20°C” and “coning and quartering,” but does not explain why different methods were chosen for specific samples. Adding rationale for the chosen methods would improve transparency.
- Comparative Analysis with Existing Literature: The manuscript could be strengthened by a more comprehensive comparison with prior studies. Including comparative data and interpretations from existing literature, especially for the reuse potential of materials like fly ash and slag, would provide a stronger foundation for the study's claims on circular economy potential.
Minor Issues
- Readability of Figures and Tables: Some tables and figures, such as Table 2 (physical characteristics) and Figure 1 (sieve analysis), are crowded and difficult to read. Consider reorganizing tables to reduce clutter and making the figures clearer by possibly splitting them into multiple panels or simplifying the data presented.
- Improved Use of References: There are instances where references to previous studies could be strengthened, particularly in the background information on specific residues (e.g., the use of fly ash in cement applications). Adding further citations would enhance the study's credibility. Al-containing industrial residues have shown potential for use in geopolymer applications, especially for improving fire resistance. However, most current research achieves similar enhancements through calcium aluminate cement (CAC) or ground granulated blast furnace slag (GGBFS). If this study could demonstrate comparable benefits by using Al-rich waste materials, it would significantly elevate its impact on the circular economy. Including the following references could enrich the discussion:
"Measurement and prediction of thermal properties of alkali-activated fly ash/slag binders at elevated temperatures"
"Effect of calcium aluminate cement on geopolymer concrete cured at ambient temperature"
These references could make the study more compelling by linking it to existing advancements in geopolymer technology."
Recommendations
This study addresses a relevant and timely topic with implications for sustainable material reuse and circular economy. Addressing the above issues, particularly the clarity in methodology, deeper statistical analysis, and practical circular economy applications, would significantly strengthen the paper’s contribution to the field.
Comments on the Quality of English Language
- A careful proofreading is recommended to address minor grammatical errors and improve flow. Simplifying some technical language would also help make the manuscript more accessible.
Author Response
Dear editor and reviewers.
Authors sincerely thanks for your time to rad the manuscript and to provide useful suggestions on how to improve it. All suggestions were taken into account at our best. All content changes are tracked, while all style changes are not tracked, because the document would be too messy and unclear. Major changes are:
1) new figure was added (Fig. 1),
2) we included 31 more references as suggested,
3) materials and methods is strengthened with additional explanations,
4) many style changes in tables and in text as suggested to improve readability.
Replies to the specific suggestions by the reviewers are listed below (marked with >>>).
We are sorry we submitted the corrections so late, but changes are quite sustainable, and we had a sick leave in meantime. We hope that the changes satisfactory reflects reviewers' suggestions. If not, then we can do our best in another revision cycle.
Looking forward to your decision.
Best regards,
Gorazd Žibret on behalf of authors
Summary of the Manuscript
This manuscript explores the characterization of Al-containing industrial residues in the East South East Europe (ESEE) region, with a focus on supporting the circular economy and the EU Green Deal. The study investigates the potential use of industrial residues, such as red mud, fly ash, and slag, as secondary raw materials in construction and other sectors, examining their physical, chemical, mineralogical, and radiological characteristics. This characterization aims to lay a foundation for their future utilization as sustainable materials aligned with circular economy principles.
Major Issues
- Methodology Clarity: The study involves a detailed analysis of various physical and chemical characteristics. However, the methodology could benefit from clearer descriptions, especially concerning sample preparation techniques, controls, and the consistency across samples from different industries. For instance, the paper mentions “drying at 20°C” and “coning and quartering,” but does not explain why different methods were chosen for specific samples. Adding rationale for the chosen methods would improve transparency.
>>> Methodological description is improved by adding additional explanations and information and also Fig. 1 containing the photographs of all samples.
- Comparative Analysis with Existing Literature: The manuscript could be strengthened by a more comprehensive comparison with prior studies. Including comparative data and interpretations from existing literature, especially for the reuse potential of materials like fly ash and slag, would provide a stronger foundation for the study's claims on circular economy potential.
>>> 30 references were added to strengthen the manuscript.
Minor Issues
- Readability of Figures and Tables: Some tables and figures, such as Table 2 (physical characteristics) and Figure 1 (sieve analysis), are crowded and difficult to read. Consider reorganizing tables to reduce clutter and making the figures clearer by possibly splitting them into multiple panels or simplifying the data presented.
>>> All tables were edited (without track changes because it would be messy). Font size was decreased to 9, column width was adjusted to increase readability. Unfortunately, we do not see how we can simplify figures without losing too much important information.
- Improved Use of References: There are instances where references to previous studies could be strengthened, particularly in the background information on specific residues (e.g., the use of fly ash in cement applications). Adding further citations would enhance the study's credibility. Al-containing industrial residues have shown potential for use in geopolymer applications, especially for improving fire resistance. However, most current research achieves similar enhancements through calcium aluminate cement (CAC) or ground granulated blast furnace slag (GGBFS). If this study could demonstrate comparable benefits by using Al-rich waste materials, it would significantly elevate its impact on the circular economy. Including the following references could enrich the discussion:
"Measurement and prediction of thermal properties of alkali-activated fly ash/slag binders at elevated temperatures"
"Effect of calcium aluminate cement on geopolymer concrete cured at ambient temperature"
These references could make the study more compelling by linking it to existing advancements in geopolymer technology."
>>> We have checked the references and included some new ones as suggested.
Recommendations
This study addresses a relevant and timely topic with implications for sustainable material reuse and circular economy. Addressing the above issues, particularly the clarity in methodology, deeper statistical analysis, and practical circular economy applications, would significantly strengthen the paper’s contribution to the field.
Comments on the Quality of English Language
- A careful proofreading is recommended to address minor grammatical errors and improve flow. Simplifying some technical language would also help make the manuscript more accessible.
>>> We have sent the manuscript for proofreading and improved English language.
Reviewer 2 Report
Comments and Suggestions for Authors
The paper “Characterization of Al-containing industrial residues in ESEE region supporting circular economy and EU Green Deal” performs a detailed characterization of different industrial by-products. However, it is recommended to include some valuation of them. Additionally, the following comments and suggestions are made
Specific comments:
Abstract:
It must include results and the most relevant conclusion of the study.
1. Introduction
Justify text (Ctrl+J)
Line 94 and 95: could be added to the requirements of SRMs: crystalline or amorphous phase).
2. Materials and Methods
Line 116. the authors used the SL2 nomenclature in this line, but did not previously define it, so it is recommended to define it.
Line 174: Use “The main chemical oxides (SiO2, Al2O3, Fe2O3, CaO, SO3, MgO, Na2O, K2O)” instead of “The main chemical oxides (SiO2, Al2O3, Fe2O3, CaO, SO3, MgO, Na2O, K2O)”,
Line 176: use P2O5 and TiO2 instead of “P2O5 and TiO2”
Line 207: use “Al2O3” instead of “Al2O3”
It is suggested to add photos of each secondary raw material.
3. Results and discussion
Table 2. improve row one of the table (all titles centered and at the same level)
Line 401. What do you mean by “while BA2 sample is depleted”.
Line 409: Use CO2 instead of CO2
Line 410: Use Al2O3 instead of Al2O3
Remember that ASTM C618 previously named the compressive strength test with the replacement of 20% as the pozzolanic activity index, but in its latest versions, it names it as a strength activity index, since exceeding 75% of the compressive strength of the reference sample does not guarantee that it has a pozzolanic effect (it may be a filler effect). It is fine to name it as meeting the chemical composition criteria to be considered as a pozzolan, but other tests are needed for reactivity.
It is recommended to add a table with the permitted limits of radioactive elements.
Line561: use CO2 instead of CO2
How could the secondary raw materials that exceed the radioactivity values be valorized?
Author Response
Dear editor and reviewers.
Authors sincerely thanks for your time to rad the manuscript and to provide useful suggestions on how to improve it. All suggestions were taken into account at our best. All content changes are tracked, while all style changes are not tracked, because the document would be too messy and unclear. Major changes are:
1) new figure was added (Fig. 1),
2) we included 31 more references as suggested,
3) materials and methods is strengthened with additional explanations,
4) many style changes in tables and in text as suggested to improve readability.
Replies to the specific suggestions by the reviewers are listed below (marked with >>>).
We are sorry we submitted the corrections so late, but changes are quite sustainable, and we had a sick leave in meantime. We hope that the changes satisfactory reflects reviewers' suggestions. If not, then we can do our best in another revision cycle.
Looking forward to your decision.
Best regards,
Gorazd Žibret on behalf of authors
Reviewer #2
The paper “Characterization of Al-containing industrial residues in ESEE region supporting circular economy and EU Green Deal” performs a detailed characterization of different industrial by-products. However, it is recommended to include some valuation of them. Additionally, the following comments and suggestions are made
Specific comments:
Abstract:
It must include results and the most relevant conclusion of the study.
>>> Main findings were added to the abstract.
- Introduction
Justify text (Ctrl+J)
>>> Text is now justified.
Line 94 and 95: could be added to the requirements of SRMs: crystalline or amorphous phase).
>>> This part was improved.
- Materials and Methods
Line 116. the authors used the SL2 nomenclature in this line, but did not previously define it, so it is recommended to define it.
>>> corrected
Line 174: Use “The main chemical oxides (SiO2, Al2O3, Fe2O3, CaO, SO3, MgO, Na2O, K2O)” instead of “The main chemical oxides (SiO2, Al2O3, Fe2O3, CaO, SO3, MgO, Na2O, K2O)”,
>>> Done
Line 176: use P2O5 and TiO2 instead of “P2O5 and TiO2”
>>> Done
Line 207: use “Al2O3” instead of “Al2O3”
>>> Done
It is suggested to add photos of each secondary raw material.
>>> Done, photos are now added as Fig. 1.
- Results and discussion
Table 2. improve row one of the table (all titles centered and at the same level)
>>> Done
Line 401. What do you mean by “while BA2 sample is depleted”.
>>> Corrected
Line 409: Use CO2 instead of CO2
>>> Done
Line 410: Use Al2O3 instead of Al2O3
>>> Done
Remember that ASTM C618 previously named the compressive strength test with the replacement of 20% as the pozzolanic activity index, but in its latest versions, it names it as a strength activity index, since exceeding 75% of the compressive strength of the reference sample does not guarantee that it has a pozzolanic effect (it may be a filler effect). It is fine to name it as meeting the chemical composition criteria to be considered as a pozzolan, but other tests are needed for reactivity.
>>> This part is removed.
It is recommended to add a table with the permitted limits of radioactive elements.
>>> "The permitted limits of radioactive elements" depend on the purpose of the material and are limited based on the estimated doses, which has already been done in the paper. An ACI of less than 1 ensures that Eall is also less than 1mSv, making these materials safe for all building uses. Therefore, exact, individual permitted limits do not exist, but the dose is estimated. If the ACI is greater than 1, doses for some more limited uses are evaluated. A sentence explaining that was added in the ch. 3.8.
Line561: use CO2 instead of CO2
>>> Done!
How could the secondary raw materials that exceed the radioactivity values be valorized?
>>> If we understand question correctly, the answer would be similar. Therefore, there are no individual values that would be allowed, but the assessment of the dose for a specific use is used to assess the health risk. Table 13 shows the procedure for samples whose ACI is greater than 1, which involves dose estimation for different situations and shows in which cases the tested materials are safe from a radiological point of view. If the comment referred to a potential situation where the annual effective dose for all situations is greater than one, it is examined whether the material can be disposed of in the environment (the criteria are defined by competent institutions in the country) and if it turns out that the material is not safe for disposal, it is reported to the competent institution (usually the radiological inspection), which passes measures on further action. An explanation was included at the end of ch. 3.8
Round 2
Reviewer 2 Report
Comments and Suggestions for Authors
The authors considered the comments and suggestions, which allowed improving the manuscript.